# The Effect of Cholesterol in MCF7 Human Breast Cancer Cells

**DOI:** 10.3390/ijms24065935

**Published:** 2023-03-21

**Authors:** Elisabetta Albi, Martina Mandarano, Samuela Cataldi, Maria Rachele Ceccarini, Federico Fiorani, Tommaso Beccari, Angelo Sidoni, Michela Codini

**Affiliations:** 1Department of Pharmaceutical Sciences, University of Perugia, 06126 Perugia, Italy; 2Division of Pathological Anatomy and Histology, Department of Experimental Medicine, School of Medicine and Surgery, University of Perugia, 06126 Perugia, Italy

**Keywords:** breast cancer, cholesterol, sphingomyelin, MCF7, cell growth

## Abstract

In the last decade, cholesterol level has been implicated in several types of cancer, including breast cancer. In the current study, we aimed to investigate the condition of lipid depletion, hypocholesterolemia or hypercholesterolemia reproduced in vitro to analyze the response of different human breast cancer cells. Thus, MCF7 as the luminal A model, MB453 as the HER2 model and MB231 as the triple-negative model were used. No effect on cell growth and viability was detected in MB453 and MB231 cells. In MCF7 cells, hypocholesterolemia (1) reduced cell growth, and Ki67 expression; (2) increased ER/PgR expression; (3) stimulated the 3-Hydroxy-3-Methylglutaryl-CoA reductase and neutral sphingomyelinase and; (4) stimulated the expression of CDKN1A gene coding cyclin-dependent kinase inhibitor 1A protein, GADD45A coding growth arrest and DNA-damage-inducible alpha protein and, PTEN gene coding phosphatase and tensin homolog. All these effects were exacerbated by the lipid-depleted condition and reversed by the hypercholesterolemic condition. The relationship between cholesterol level and sphingomyelin metabolism was demonstrated. In summary, our data suggest that cholesterol levels should be controlled in luminal A breast cancer.

## 1. Introduction

### 1.1. Breast Cancer

Scientific research aimed to study the pathogenetic mechanisms and clinical implications of cancer has always focused, and still is strongly concentrated on, breast cancer. Certainly, scientific progress has been remarkable in recent years, but, despite this, breast cancer remains the most common cause of cancer in women and affects increasingly younger women [1]. Mammography and ultrasound were recognized as the most effective diagnostic techniques for women with non-dense and dense breast tissue, respectively [2]. Additionally, breast self-examination, magnetic resonance imaging, high-definition digital mammography, and contrast-enhanced spectral mammography, optical mammography, radiothermometry mammography, scintimammography, positron emission tomography helped to early detect tumors or lesions predisposing to tumors [3]. Thus, alongside progress in diagnosis, genomic studies and the identification of new biomarkers have made it possible to implement effective precision medicine [4]. Such clinical direction with consequent combinatorial therapeutic strategies was derived from the scientific awareness that breast cancer was not a single disease. In fact, the disease could be characterized by intratumoral heterogeneity with even different stem cells leading to the coexistence of multiple cancer subtypes within the tumor itself [5]. The histological classification of breast tumors has occurred based on specific receptor expressions such as the estrogen receptor (ER), the progesterone receptor (PgR) and the human epidermal growth factor receptor 2 (HER2) in addition to the percentage of the proliferating index (Ki67) [6]. Classically, different subtypes of breast cancer were considered: luminal A, luminal B and triple-negative basal-like (TNBC) [6]. Luminal A is ER+ and/or PgR+ and HER2- [7]. Luminal B is further classified into luminal B HER2- and luminal B HER2+. Of note, the luminal B HER2- is ER+ and/or PgR+ and HER2-, and the luminal B HER2+ is ER +and/or PgR+ and HER2+ [7]. Non-luminal A and B HER2- are hormone negative and have a worse prognosis than luminal types. Finally, TNBC is ER-, PgR-, and HER2- and has the worst prognosis of all subgroups [7]. 

Additionally, a triple negative claudin-low and HER2 (ER-, PgR- and HER2-) subtype was also described [8]. Genomic and/or transcriptomic analyses led to the identification of 10 breast cancer subtypes based on integrated clusters, four ER-negative and six triple-negative breast cancer subtypes [5]. Interestingly, the histological type classification of breast cancer is fundamental for both prognosis and choice of therapeutic treatment.

### 1.2. Cholesterol in Cancer

Cholesterol (Chol) mostly exerted opposite functions in physiological and pathological conditions as a source of hormones and vitamins, acting as atherogenic in the cardiovascular system, and acting as a pro-growth agent in cancer [9]. Hypercholesterolemia facilitated the entry of Chol into cancer cells [10] due to the overexpression of the LDL receptor in these cells [11]. Moreover, metabolic reprogramming in transformed cells was described [12]. Thus, the upregulation of enzymes for Chol synthesis was demonstrated in cancer cells [13,14]. Mechanistically, Chol entered the cancer cells and appeared to disorganize lipid rafts in the cell membrane [15] and nuclear membrane [16]. The deregulation of nuclear membrane lipid rafts changed the chromatin function and the response to drugs [17]. In support, Chol stimulated DNA synthesis [18]. Additionally, Chol enrichment could enhance tumor microenvironmental adaptability [19] and mediate immune and epigenetic mechanisms [20], reinforcing tumor incidence and progression. Chol entry into tumor cells increased as the disease worsened, resulting in severe hypocholesterolemia [21,22]. In liquid tumors, severe hypocholesterolemia was accompanied by severe hypophospholipemia and high levels of antiphospholipid antibodies [23].

### 1.3. Cholesterol–Sphingomyelin Relationship in Cancer

Chol has been shown to link sphingomyelin (SM) and form lipid rafts [24]. In cancer cells, the number of lipid rafts increases due to Chol synthesis enzymes upregulation [25]. Conversely, the efflux of Chol from lipid rafts by specific transporters induced a phenotypic reprogramming of macrophages to become more sensitive to protumor signals [26]. The amount of Chol that became free to leave the raft was strictly dependent on the SM metabolism. In fact, sphingomyelinase (SMase) degrades SM in ceramide and phosphocholine, allowing the release of Chol [27]. The produced ceramide was converted to sphingosine, which was converted to sphingosine-1-phosphate by sphingosine kinase (SphK). If SphK, especially type 1, moved within the microdomain, it had an oncogenic effect; if it localized outside the microdomain, it inhibited cell proliferation [28]. This is relevant considering that SphK was implicated in several cancers, including colon, lung, kidney, and brain cancers [29]. Interestingly, altered sphingolipid levels were a frequent phenomenon during cancer progression, so much so that it was considered a possible marker to distinguish the luminal form from the TNBC of breast cancer [30]. Thus, a close correlation between Chol and SM metabolism resulted in altered cancer [16]. There are currently no data on the comparative effect of HypoChol and HyperChol on the same histological cell type.

The present study tested the effects of two different Chol conditions reproducing hypocholesterolemia (HypoChol) and hypercholesterolemia (HyperChol) in MCF7, MB453 and MB231 breast cancer cells. No differences were detected in MB453 and MB231 cells, while cell growth of MCF7 was influenced by Chol concentration. Therefore, the second part of the study aimed to investigate the possible metabolic mechanisms involved in the change in the Chol level. Notably, the effect of SM metabolism was investigated by studying neutral SMase 3 (nSMase3), encoded by the SMPD4 gene, which is deregulated in human cancer [31].

## 2. Results

### 2.1. Cholesterol Does Not Affect MCF7, MB453 and MB231 Breast Cancer Cells in the Same Way 

We first sought to characterize differences in terms of hormone receptor expression, cell morphology and growth on MCF7, MB453 and MB231 breast cancer cells. As expected, only MCF7 cells expressed ER and PgR, according to Holliday and Speirs (Appendix A) [8]. 

MCF7 appeared with an ovoid and incurved cell body with strong cell–cell adhesion, MB453 were round with small aggregates, and MB231 had a fusiform morphology and showed cellular dispersion (Appendix A). MB453 cells undergo cell growth faster and MB231 even faster than MCF7, as could be expected due to the different tumor aggressiveness, as reported in the introduction (Appendix A).

We analyzed the morphology, cell number and viability of MCF7, MB453 and MB231 breast cancer cells in our experimental model CTR, LD, HypoChol and HyperChol (see methods). After 48 h, cell culture differences in morphology and cell number were evident only in MCF7 cells (Figure 1a,b). Indeed, in these cells, LD to a greater extent and hypoChol to a lesser extent reduced cell growth. Conversely, the HyperChol had a stimulating effect with a greater number of elongated cells. No changes in cell number were statistically significant in MB453 and MB231 cells. Furthermore, cell viability was unchanged in a Chol-dependent manner in all cell types (Figure 1a,b).

To evaluate whether the endogenous Chol content was different in the three cell lines and whether the variation of the exogenous Chol might influence the endogenous one, the Chol dosage was performed in all samples. The results showed that MB453 and MB231 cells were richer in Chol content than MCF7. Moreover, the treatments of LD, HypoChol and HyperChol conditions affected the Chol content only in MCF7 cells (Figure 2).

### 2.2. Involvement of Cholesterol in Several Biological Phenomena of MCF7 Cells 

Since Chol level has emerged as a modulator of cell morphology and growth only in MCF7 cells without any effect in other cell lines, we aimed to explore whether this prediction translated into specific cellular activities. Thus, MCF7 cells in LD, HypoChol and HyperChol conditions were tested for Ki67, ER and PgR expression. LD dramatically reduced the Ki67 and strongly increased ER and PgR. HypoChol induced similar cell biological changes with a more moderated effect, and HyperChol had the opposite effect (Figure 3a,b). Of note, LD and HypoChol were responsible for 66% and 32% Ki67 expression reduction, respectively. ER was 27% and 13% upregulated, and PgR was 33% and 27.7% upregulated by LD and HypoChol conditions, respectively. Conversely, HyperChol induced 12.5% upregulation of Ki67 and down-regulated ER (91.4%) and PgR (30.6%) (Figure 3b).

Next, we attempted to establish whether the variation of exogenous Chol could influence its endogenous synthesis. To that end, we evaluated the key enzyme for Chol synthesis, the 3-Hydroxy-3-Methylglutaryl-CoA reductase (HMGCR). The gene expression was overexpressed in LD and HypoChol conditions and down-expressed in HyperChol conditions (Figure 4a). Accordingly, the protein expression was upregulated in LD and down-regulated in HyperChol. However, the HypoChol condition did not induce a translation of gene-protein expression since the value of protein expression was similar to the control (Figure 4b,c).

Most importantly, Chol level influenced the expression of gene coding cell cycle regulation proteins, growth arrest and tumor suppression (Figure 5). In fact, LD and HypoChol upregulated *CDKN1A* gene coding cyclin-dependent kinase inhibitor 1A protein, *GADD45A* coding growth arrest and DNA-damage-inducible alpha protein and *PTEN gene* coding phosphatase and tensin homolog, an oncosuppressor protein. These genes were down-expressed in the HyperChol condition, which instead stimulated overexpression of *CCND1* coding cyclin D1 (Figure 5).

### 2.3. Relationship between Chol Level and Sphingomyelin Metabolism in MCF7 Cells

In the last several years, the link between Chol-SM in cancer has been widely described, as reported in the “Introduction”. Thus, we analyzed the gene and protein expression of nSMase3 in relation to a different level of Chol in the culture medium. MCF7 cells cultured in LD and/or HypoChol conditions overexpressed the SMPD4 gene (Figure 6a) coding for nSMase3, as reported above. The nSMase3 protein expression was upregulated with LD and down-regulated with HyperChol (Figure 6b,d). Considering the enzyme activity of total nSMases expressed in relation to mg total protein, a strong increase under LD conditions appeared (Figure 6c). However, it must be considered that the activity refers to the total nSMases since there are currently no specific kits for nSMase 3 activity. Therefore, this result is only indicative. 

Next, the effects of nSMase3 on cell growth, cell viability, Ki67, ER and PgR were investigated on MCF7 cells. The number of cells after 48 h of culture was significantly reduced with increasing concentration of nSMase3, and cell viability was consistently above 80% (Figure 7a).

The higher concentration of 100nM was responsible for the greater reduction in the number of cells that had high viability in any case. Thus, we used this concentration to carry out the microscopy and immunohistochemical study. The microscopic analysis confirmed the reduction of cell growth after 48 h of culture. Immunohistochemical analysis revealed overexpression of Ki67 and no significant changes in ER and PgR expression.

## 3. Discussion

In the present study, we have defined a differentiative response to the Chol level of MCF7 cells with respect to MB435 and MB231 cells. Under the same experimental conditions, only MCF7 responded to the change in Chol content. The possibility that by varying the experimental conditions, MB453 and MB231 could be influenced by the level of exogenous Chol cannot be excluded. Future studies may clarify this point. The results indicated that the changes in the culture medium Chol levels affected only human breast cancer cells that express ER and PgR and therefore respond to hormone therapy [32]. 

Chol has been known for a long time to increase cell proliferation [33]. Despite compelling evidence across multiple studies that no effects in colorectal, lung, prostate, and breast cancer incidence were reported [34], some clinical studies showed that statins might reduce the risk of breast cancer recurrence [35]. However, a recent systematic review on the association between serum lipid profile and risk of breast cancer incidence concluded that the total Chol and LDL had no role in breast cancer incidence [36]. In contrast, in a clinical study conducted on 170 patients in China by Sun et al. [37], circulating levels of total Chol and LDL were markedly higher in patients with triple-negative and HER2-positive breast cancer than in luminal A- and B-negative patients. Thus, clinical data on the association between Chol levels and breast cancer are conflicting. Our results indicated that the content of Chol was higher in MB453 and MB231 than in MCF7 cells, indicating that breast cancer cells unresponsive to hormone therapy in the absence of ER and PgR [8] accumulate Chol. It is possible to hypothesize that intracellular Chol level might be correlated with the response to hormone therapy. The level of exogenous Chol influenced the intracellular Chol level and simultaneously cell growth only in MCF7 cells. The data suggested that reducing Chol levels pharmacologically might be important in luminal A tumors corresponding to the MCF7 cell line. For instance, in MCF7 cells, LD and HypoChol conditions down-expressed Ki67 and overexpressed ER. Our data support previous evidence demonstrating that in ER+ breast cancer, exogenous Chol may be relevant to the tumor microenvironment and its metabolite, 27-hydroxyChol, may modulate ER activity [38]. Therefore, our data demonstrate the effect of exogenous Chol on the behavior of breast cancer cells. Furthermore, several authors focused attention on the role of endogenous Chol in breast cancer cells, especially Chol located in membrane lipid rafts. In fact, endogenous increased Chol synthesis is involved in tumor sphere formation and invasion [39]. The invasion is particularly due to the Chol-enriched domains due to their control in invadopodia and the degradation of the extracellular matrix [40]. Shi et al. reported the involvement of Chol-enriched membrane micro-domains deficiency in autophagosome biogenesis and doxorubicin resistance in breast tumors [41]. Moreover, partial membrane Chol depletion decreases invasion of the malignant cell lines [40]. 

It has recently been shown that MCF7 cell response to anticancer drugs was influenced by the sphingolipid profile [42]. In the current study, we found that changes in Chol levels influenced nSMase3 expression. A very low level of Chol in the LD condition increased nSMase3 gene and protein expression, and the effect was reversed by the HyperChol condition. The HypoChol condition stimulated SMPD4 gene expression, which did not result in an upregulation of the nSMase3 enzyme. Therefore, it would probably be necessary to prolong the experimental procedure in time to obtain the effect of the LD condition. However, low and high levels of Chol had opposite effects on the enzyme for SM metabolism. It is possible that the very low level of Chol in the LD condition increased nSMase3 to produce ceramide, which was known to stimulate breast cancer cell apoptosis [43]. Conversely, HyperChol could reduce nSMase3 to increase SM content. This would probably lead to an increase in lipid rafts which were implicated in cancer [16]. Additionally, treatment of the cells with exogenous nSMase induced a reduction in cell number and Ki67 expression without affecting ER and PgR expression, suggesting that HypoChol might stimulate nSMase3 to reduce cell growth [24]. Future research on breast cancer luminal A could be aimed at (1) clarifying the mechanism of action of Chol on gene transcription; (2) defining the role of the Chol-induced SM metabolism change on the cell cycle.

In conclusion, our data show the role of Chol on luminal A breast cancer and highlight the possible involvement of nSMase3. 

## 4. Materials and Methods

### 4.1. Chemicals

MDA-MCF7 and MDA-MB231 cell lines were from Elabscience Biotechnology (Houston, Texas, USA), and MDA-MD453 was from ATCC (ATCC^®^ HTB-131, Manassas, VA, USA). Cell culture medium (Dulbecco’s modified minimum essential medium, DMEM), fetal bovine serum (FBS), penicillin G, streptomycin, glutamine, and sodium pyruvate were from GIBCO Invitrogen (Carlsbad, CA, USA). FBS lipid-depleted was from Biowest (Nuaillé, France). Human LDL was from Millipore (Burlington, Massachusetts, USA). Anti-nSMase3, anti-β-tubulin, anti-HMGCR and horseradish peroxidase-conjugated goat anti-rabbit secondary antibodies were from Abcam (Cambridge, UK). ER antibody, clone 6F11, was from Leica Biosystems, Newcastle Upon Tyne, United Kingdom. PgR antibody, clone 16, was from Leica Biosystems, Newcastle Upon Tyne, United Kingdom. Ki-67 antibody clone MIB1 was from Agilent Dako, Glostrup, Denmark. 3-[4,5-dimethyl-2-thiazolyl]-2,5-diphenyl-2-tetrazoliumbromide (MTT), Chol standard, nSMase and were from Sigma-Aldrich (St. Louis, MO, USA). TaqMan SNP Genotyping Assay and Reverse Transcription kit were purchased from Applied Biosystems (Foster City, CA, USA). RNAqueous^®^-4PCR kit was from Ambion Inc. (Austin, TX, USA). SDS-PAGE molecular weight standards were purchased from Nzythech (Lisboa, Portugal). Chemiluminescence kits were purchased from Amersham (Rainham, Essex, UK).

### 4.2. Cell Culture and Treatments

MCF7 as luminal A model (ER+, PgR+/−, HER2), MB435 as HER2 basal model (ER^−^, PgR^−^, HER2^+)^ and MB231 as triple-negative claudin-low (ER-, PgR-, HER2-) were used [8]. Cells were grown in DMEM supplemented with 10% heat-inactivated FBS, 100 IU/mL penicillin/streptomicyn, and 200 mM of L-glutammine. Cells were maintained at 37 °C in a saturating humidity atmosphere containing 95% air and 5% CO_2_. To assess differences in cell growth of MCF7, MB435 and MB231 cell lines, the cells were counted at different times of culture (0 h, 24 h, 48 h, 72 h, 96 h, 168 h).

For experiments, four lots of the MCF7, MB453, and MB321 cells were prepared considering the content of Chol in the normal fetal bovine serum (FBS) and lipid-depleted (LD) FBS to induce a HypoChol and HyperChol condition. FBS contained 378 ± 3mg/dL of phospholipids (PLs) and 32 ± 4 mg/dL of Chol, corresponding to approximately 150 mg/dL of Chol in human serum, according to Haylett and Moore (2002) [44]. In LD, FBS PLs were 108 ± 4 mg/dL, and Chol was 3.0 ± 0.2 mg/dL, corresponding to approximately 15 mg/dL Chol in human serum. Thus, we added 10mg/dL LDL, corresponding to approximately 47 mg/dL, to LD FBS to obtain a 62 mg/dL HypoChol condition [2]. Then, we added 20 mg/dL LDL, corresponding to approximately 94 mg/dL, to FBS to obtain a 244 mg/dL HyperChol condition. Therefore, we had 4 samples: the control sample (CTR) cultured with 10% normal FBS; lipid-depleted sample (LD) cultured with 10% FBS lipid-depleted; hypocholesterolemia (HypoChol) cultured with 10% FBS lipid-depleted in the presence of low-density lipoproteins (LDL) (10 mg/dLPBS); hypercholesterolemia (HyperChol) cultured with 10% normal FBS in the presence of LDL (20 mg/dL PBS)—differences in cell number after treatments were analyzed. To test the effect of nSMase on MCF7 cells, increasing concentration of the enzyme was used as reported below. 

### 4.3. Cell Viability

MTT assay was used to test cell viability, as previously reported [45]. MCF7, MB435 and MB231 (CTR, LD, HypoChol and HyperChol) were seeded into 96-well plates at a density of 1 × 10^4^ cells/well with DMEM complete culture medium. After 24 h, culture medium was replaced with respective culture medium with or without LDL, as reported above for each sample, and the cells were incubated for 24 h. Samples with 1% or 4% DMSO were added as controls due to their cytotoxic effect [46,47,48]. Then, MTT reagent was dissolved in PBS 1x and added to the culture at 0.5 mg/mL final concentration. After 3 h of incubation at 37 °C, the supernatant was carefully removed, and formazan salt crystals were dissolved in 200 µL DMSO that was added to each well. The absorbance (OD) values were measured spectrophotometrically at 540 nm using an automatic microplate reader (Eliza MAT 2000, DRG Instruments, GmbH, Marburg, Germany). Each experiment was performed two times in triplicate, and the results were expressed as a percentage relative to the control cells.

### 4.4. Cell Morphology

MCF7, MB435 and MB231 were cultured for 48 h, as reported above. The observations were performed by using inverted microscopy EUROMEX FE 2935 (ED Arnhem, Netherlands) equipped with a CMEX 5000 camera system (20× or 40× magnification) and the morphometric analysis was performed by using ImageFocus software, as reported above [49].

### 4.5. Cholesterol Analysis

Lipids were extracted, and total phospholipids (PLs) and Chol were measured, as previously reported [50]. Briefly, lipids were extracted with 20 volumes of chloroform/methanol (2:1 vol/vol for culture medium and the suspended pellet of the cells). After 2 h, the organic phase was filtered with 42.5 mm diameter Whatman filter paper from Sigma-Aldrich (St. Louis, MO, USA), washed with 0.2 volume of 0.5% NaCl and kept overnight in a separating funnel at 4 °C. After separation, the aqueous phase was extracted again with chloroform and methanol twice and solvent containing NaCl. Lipids were also present in the second and third extraction, as previously reported [51]. The total amount of phospholipids (PLs) was determined by measuring inorganic phosphorus. Chol was separated by thin-layer chromatography (TLC), using hexane/diethyl ether/formic acid, 80:20:2 by volume, as solvent. The spot of Chol was localized using the Chol standard as reference and showed Rf of 0.24; the quantification was made according to Rudel and Morris [52]. 

### 4.6. Immunocytochemistry

MCF7 cells were cultured for 48 h, then centrifuged at 1000× *g* for 10 min and treated as previously reported [53]. The pellets were fixed in 10% neutral phosphate-buffered formaldehyde solution for 24 h. Then, the cytoinclusion technique by the Cellient^®^ Automated Cell Block System (Hologic, Mississauga, ON, Canada) that rapidly creates a paraffin-embedded cell block was used, as previously reported. The inclusion was sectioned into 4 µm thick sections. Bond Dewax solution was used to remove paraffin from sections before rehydration and immunostaining on the Bond automated system (Leica Biosystems Newcastle, Ltd., Newcastle upon Tyne, UK). Immunostaining for ER, PgR and Ki-67 detection was performed using specific antibodies and Bond Polymer Refine Detection—Leica Biosystems (Newcastle, Ltd., Newcastle upon Tyne, UK). The observations were performed using inverted microscopy EUROMEX FE 2935 (ED Arnhem, The Netherlands) equipped with a CMEX 5000 camera system (40× magnification). The intensity of immunostaining was evaluated. The findings were classified as no reactive, low positive, medium positive, and strong positive cells. Only the strong positive cells were considered for quantification, as previously reported [54].

### 4.7. Reverse Transcription Quantitative PCR (RTqPCR)

Total RNA was extracted from MCF7 cells cultured for 24 h, as reported above, using RNAqueous-4PCR kit, as previously described [55]. Samples were treated with RNase-free DNase to prevent amplification of genomic DNA possibly present. Samples were dissolved in RNase-free water, and total RNA amount was quantified by measuring the absorbance at 260 nm (A260). The purity of RNA was evaluated by using the A260/A280 ratio. A260/A230 ratio was also used to indicate chemical contaminants in nucleic acids. The extracted RNA was immediately frozen and maintained at −80 °C. Before cDNA synthesis, the integrity of RNA was evaluated by electrophoresis in TAE 1.2% agarose gel. cDNA was synthesized using 1μg total RNA for all samples by High-Capacity cDNA Reverse Transcription kit under the following conditions: 50 °C for 2 min, 95 °C for 10 min, 95 °C for 15 s and 60 °C for 1 min, for a total of 40 cycles. The following target genes were investigated: HMGCR (Hs00168352_m1), SMPD4 (Hs04187047_g1), CCND1 (HS00765553), CDKN1A (Hs_00355782_m1), GADD45A (Hs_00169255_m1), PTEN (Hs_02621230_s1). GAPDH (Hs99999905_m1) and 18S rRNA (S18, Hs99999901_s1) were used as housekeeping genes. mRNA relative expression levels were calculated as 2^−ΔΔCt^, comparing the results of the treated samples with control sample [56].

### 4.8. Protein Concentration and Western Blotting

Protein concentration was measured as previously reported with modifications [55]. Proteins (40 μg) were submitted to 10% SDS (sodium dodecyl sulfate)-polyacrylamide gel electrophoresis at 200 V for 60 min. Briefly, proteins were transferred onto 0.45 μm cellulose nitrate strips membrane (Sartorius StedimBiotech S.A.) in transfer buffer for 1 h at 100 V at 4 °C [55]. The membranes were blocked for 30 min with 5% no-fat dry milk in PBS (pH 7.4) and incubated overnight at 4 °C with HMGCR or nSMase3 specific antibodies. Anti-βtubulin antibody was used to normalize the data. The blots were treated with HRP-conjugated secondary antibodies for 90 min. Band detection was performed using an enhanced chemilumiescence kit from Amersham Pharmacia Biotech (Rainham, Essex, UK). A densitometric analysis was performed by Chemidoc Imagequant LAS500–Ge Healthcare-Life Science (Milano, Italy) [54].

### 4.9. Neutral Sphingomyelinase Activity Assay

nSMase activity was assayed as previously reported [56]. Cells were suspended in 0.1% NP-40 detergent in PBS, sonicated for 30 s on ice at 20 watts, kept on ice for 30 min and centrifuged at 16,000× *g* for 10 min. The supernatants were used for nSMase assay. The enzyme activity was assayed in 60 µg proteins/10 µL Tris-MgCl_2_, pH 7.4, using Amplex Red Sphingomyelinase assay kit (Invitrogen, Monza, Italy) according to the manufacturing instructions. The fluorescence was measured with FLUOstar Optima fluorimeter (BMG Labtech, Offemburg, Germany), using the filter set of 360 nm excitation and 460 nm emission.

### 4.10. Statistical Analysis

Three independent experiments performed in duplicate were carried out for each analysis. Data are expressed as mean ± SD; Student’s *t*-test was used for statistical analysis; each LD, HypoChol and HyperChol sample was versus the control sample (CTR). To compare the effect of HyperChol versus LD and/or HypoChol, ANOVA test was used.

## Figures and Tables

**Figure 1 ijms-24-05935-f001:**
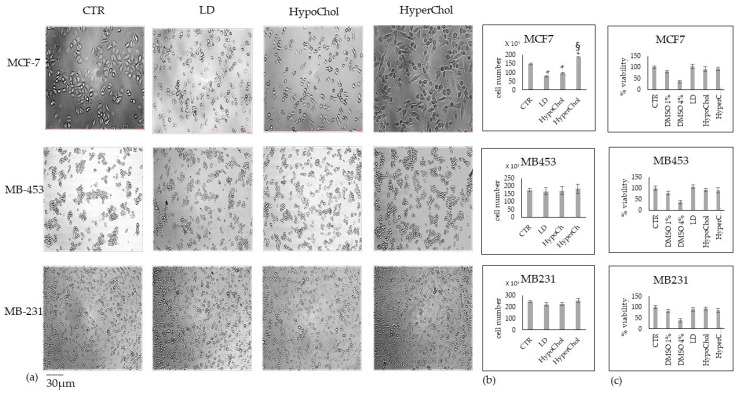
Effect of lipid-depleted (LD), low level of cholesterol (HypoChol) and high level of cholesterol (HyperChol) in MCF7, MB453, MB231 cells after 48 h of culture. (**a**) cell morphology (40× magnification); (**b**) cell number; (**c**) cell viability. Data are expressed as the mean ± SD of 3 independent experiments performed in duplicate. * *p* < 0.05 versus CTR; ^§^
*p* < 0.05 versus LD and HypoChol.

**Figure 2 ijms-24-05935-f002:**
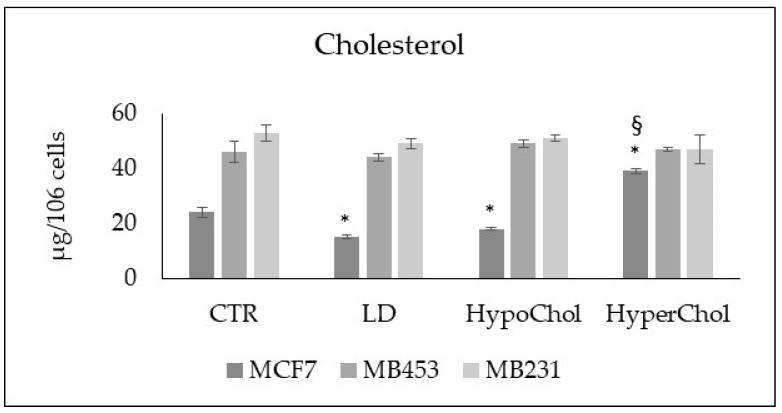
Cholesterol content in MCF7, MB435 and MB231 cells. CTR—control sample; LD—cells cultured in lipid-depleted conditions; HypoChol—cells cultured in hypoChol conditions; HyperChol—cell cultured in HyperChol conditions. Data are expressed as the mean ± SD of 3 independent experiments performed in duplicate. * *p* < 0.05 versus CTR; ^§^
*p* < 0.05 versus LD and HypoChol.

**Figure 3 ijms-24-05935-f003:**
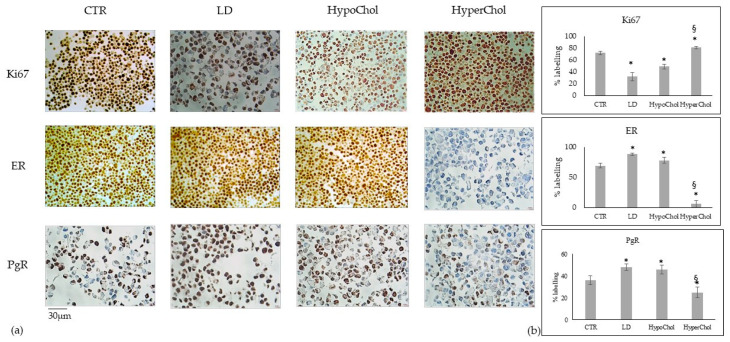
Immunocytochemistry analysis of Ki67, ER and PgR in MCF7 cells cultured in lipid-depleted (LD), low level of cholesterol (HypoChol) and high level of cholesterol (HyperChol) conditions. (**a**) positive immunostaining; (**b**) analysis of positive cells expressed in percent. Data are expressed as the mean ± SD of 2 independent experiments performed in duplicate. * *p* < 0.05 versus CTR; ^§^
*p* < 0.05 versus LD and HypoChol.

**Figure 4 ijms-24-05935-f004:**
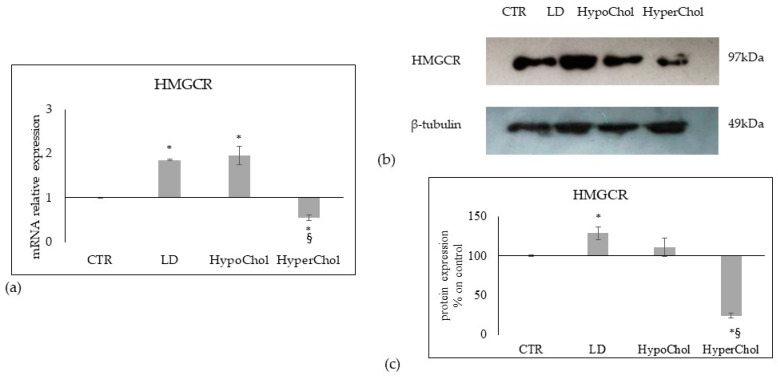
Effect of lipid-depleted (LD), low level of cholesterol (HypoChol) and high level of cholesterol (HyperChol) on key enzymes for cholesterol synthesis, 3-Hydroxy-3-Methylglutaryl-CoA reductase (HMGCR), in MCF7 cells. (**a**) gene expression of HMGCR, GAPDH and 18S rRNA were used as housekeeping genes. mRNA relative expression levels were calculated as 2^−ΔΔCt^, comparing the results of the treated samples with control sample; (**b**) Western blotting panel of HMGCR and beta-tubulin as reference; (**c**) densitometric analysis, the values were normalized with beta-tubulin protein and were expressed as percentage of control sample. Data are expressed as the mean ± SD of 3 independent experiments performed in duplicate. * *p* < 0.05 versus CTR; ^§^
*p* < 0.05 versus LD and HypoChol.

**Figure 5 ijms-24-05935-f005:**
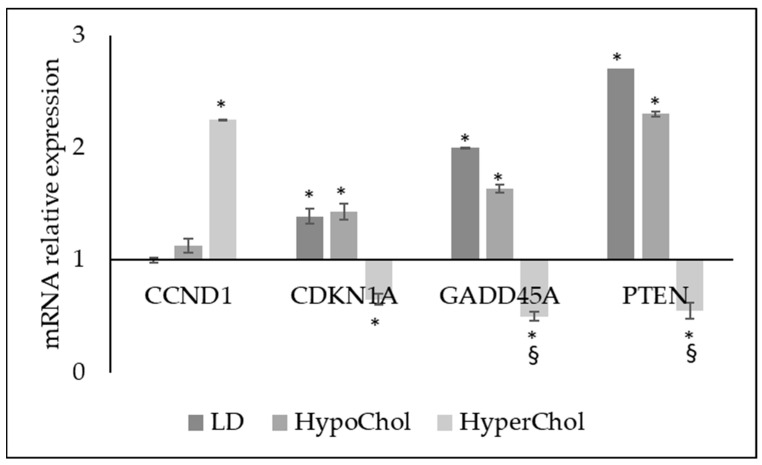
Gene expression of *CCND1* coding cyclin D1 protein, *CDKN1A* gene coding cyclin-dependent kinase inhibitor 1A protein, *GADD45A* coding growth arrest and DNA-damage-inducible alpha protein and *PTEN gene* coding phosphatase and tensin homolog protein. GAPDH and 18S rRNA were used as housekeeping genes. mRNA relative expression levels were calculated as 2^−ΔΔCt^, comparing the results of the treated samples with control sample equal to 1, the origin of axes. Data are expressed as the mean ± SD of 3 independent experiments performed in duplicate. * *p* < 0.05 versus CTR; ^§^
*p* < 0.05 versus LD and HypoChol.

**Figure 6 ijms-24-05935-f006:**
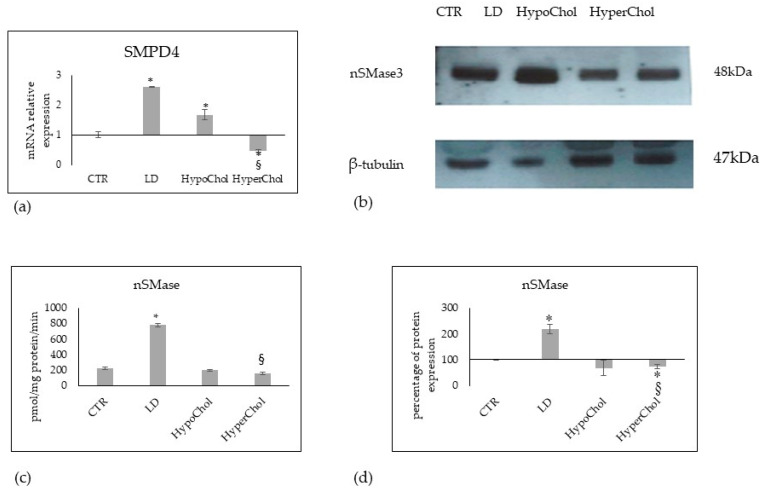
Effect of lipid-depleted (LD), low level of cholesterol (HypoChol) and high level of cholesterol (HyperChol) on neutral sphingomyelinase3 (nSMase3) in MCF7 cells. (**a**) gene expression of SMPD4 coding for nSMase3, GAPDH and 18S rRNA were used as housekeeping genes. mRNA relative expression levels were calculated as 2^−ΔΔCt^, comparing the results of the treated samples with control sample; (**b**) Western blotting panel of nSMase3 and beta-tubulin as reference; (**c**) enzyme activity; (**d**) densitometric analysis of Western blotting, the values were normalized with beta-tubulin protein and were expressed as percentage of control sample. Data are expressed as the mean ± SD of 3 independent experiments performed in duplicate. * *p* < 0.05 versus CTR; ^§^
*p* < 0.05 versus LD and HypoChol for (**a**) and versus LD for (**c**,**d**).

**Figure 7 ijms-24-05935-f007:**
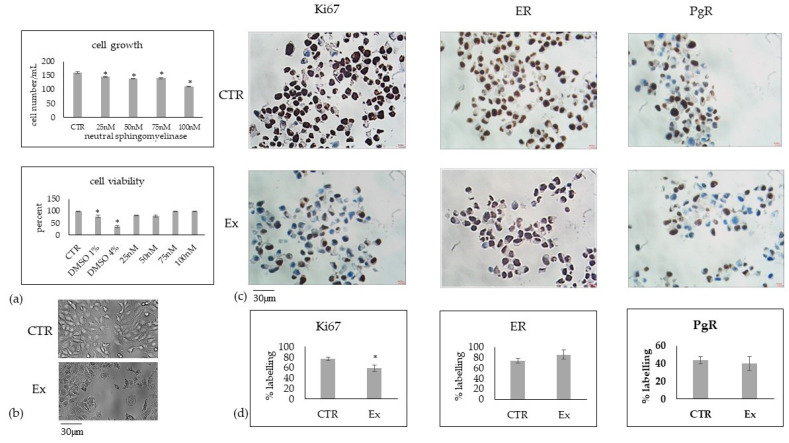
Effect of increasing concentration of neutral sphingomyelinase on MCF7 cells. (**a**) cell growth and viability; (**b**) microscopy analysis (40× magnification); (**c**) immunohistochemistry analysis of Ki67, ER and PgR; (**d**) analysis of positive cells expressed in percent. CTR—control sample; Ex—cell treated with 100nM nSMase. Data are expressed as the mean ± SD of 2 independent experiments performed in duplicate. * *p* < 0.05 versus CTR.

## Data Availability

Not available.

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
