# Peer review of "The Effect of Cholesterol in MCF7 Human Breast Cancer Cells"

_ijms, 2023, doi:10.3390/ijms24065935_

Round 1

Reviewer 1 Report (Previous Reviewer 1)

Authors partly answered to some of my comments but this is not sufficient and some new data weaken conclusions on some parts of the paper. Specifically:

1. As recommended, authors have now compared the three cell lines for their basic cholesterol content and for the effect of the different modulation conditions. They found that MB453 and MB231 cells were more enriched in chol than MCF7. Moreover, LD, hypoCho and hyperChol conditions treatments affected the Chol content only in MCF7 cells. This is problematic because it suggests that the conditions used to modulate chol in MB453 and MB231 cell lines were not appropriate to conclude on the absence of chol implication in MB453 and MB231 cell growth and viability. Indeed, an absence of effect can simply result from the fact that the chol content was simply not modulated. Authors did not comment at all on this problem.

In those conditions, authors must either remove all the data related to MB453 and MB231 cell lines and at the same time revise their conclusions on MCF7 only, without making any comparison with other cell lines; or perform additional experiments aiming at modulating chol content by conditions/approaches that can effectively reduce/increase cholesterol content in those 2 cell lines.

2. In their abstract authors mentioned that ‘In the last decade cholesterol level has been implicated in several types of cancer. However, the relationship between cholesterol content and distribution and breast cancer is poorly explored’. This is not true. Plenty of papers have explored this topic, including recent research studies, including Shi et al, Molecular Therapy 2021, Maja et al, Cell Mol Life Sci 2022, Kim et al, Biomedicines 2022; and several reviews Nelson et al, 2015, Maja and Tyteca, 2022. It is needed that authors discuss those papers in light of their own data.

3. In the previous version of the manuscript, the authors mentioned that their cell lines are more or less malignant but this was not objectified by criteria for measuring aggressiveness. In their revised version, authors did still not measure invasion of their 3 cell lines and only removed the term ‘malignant’.

Author Response

Reviewer 2 Report (Previous Reviewer 2)

The authors have addressed the questions in the previous iteration. They have replaced the images in multiple figures and provided explanations to the concerns. The authors have introduced two supplementary figures in the revised version but they are present in the main manuscript as opposed to a separate supplemental file. These need to be moved to a different file.

Author Response

Reviewer 3 Report (Previous Reviewer 3)

The authors have taken into account the suggestions and corrections and the manuscript has been well improved.

Author Response

Reviewer 4 Report (Previous Reviewer 4)

The paper was improved significantly. Please revise references from number 32 onwards; there is no coincidence with the text.

Round 2

Reviewer 1 Report (Previous Reviewer 1)

Can be accepted for publication even I am not convinced by the answers provided by the authors

This manuscript is a resubmission of an earlier submission. The following is a list of the peer review reports and author responses from that submission.

Round 1

Reviewer 1 Report

The goal of the study was to evaluate the effects of cholesterol modulation in 3 breast cancer cells. Although the question is relevant, this is not really new and the study is not convincing both in terms of models, readouts and interpretation of the data.

Major comments:

1) the authors compare three lines in 2D; they are not isogenic, are cultured in 2D only (no spheroids or organoids) and are only 3 in number. When we know that there are about twenty different subtypes of breast cancer, this is a bit weak.

2) the authors do not seem to have characterized their lines to verify that they express specific molecular markers of the different subtypes such as ER, PR, HER2, claudin, cytokeratins, a.o.

3) the authors mention that their lines are more or less malignant but this is not objectified by criteria for measuring aggressiveness.

4) the study is centered on cholesterol and the use of methods aimed at modulating intracellular cholesterol. However, the authors did not compare the 3 lines for their basic cholesterol content nor for the effect of the different modulation conditions.

5) the conclusion that chol plays a role only in the least aggressive cells is too preliminar and not supported by sufficiently solid observations.

6) the material and methods are not sufficiently described.

Minor points:

1) the text is filled with spelling and syntax errors

2) the nomenclature of the cell lines is inconsistent throughout the paper

3) images are blurry

4) the graphs are much too small and difficult to read

5) the choice of DMSO 1% and 4% is more than surprising and not justified

6) immunohistochemistry should be replaced by immunocytochemistry

7) what type is nSMase

8) method for quantification of immunocytochemistry images is not described

Reviewer 2 Report

The Effect Of Cholesterol In Mcf7 Human Breast Cancer Cells

The authors have shown the effects of differing lipid concentration on MCF7 (Luminal A) cell line. They show changes in cell number and cell viability upon providing altered lipid concentrations including a Lipid Depleted (LD) condition. They also show the involvement of nSMase in these cells upon these treatments.

In Figure 3a, under LD treated and stained for PgR, the cell density does not match the other figures and is not representative of the data presented in 3b. The authors should replace this image with one representing similar cell densities.

In Figure 4a, 4c the treatments should be compared to their respective control groups similar to Figure 3b. 

In Figure 6, the authors show an increase in nSMase in both RNA and protein levels upon LD treatment. However, they have also shown that nSMase activity remained unchanged. This contrasting effect needs to be addressed as the following experiment depicted in Figure 7 relies on the action of nSMase on MCF7 cells. 

If nSMase activity is unaffected by different lipid concentrations, then the authors should explain as to how the cellular effects are observed under nSMase treatment.

In Figure 7b, 7c and 7d, the authors refer to treated samples as “Ex”. This should be changed to correspond to the dosage used in the experiment. Furthermore, cellular densities need to be comparable to the control image. This should be changed to accurately represent the quantification in Figure 7d.

The authors conclude that a dietary approach with lower cholesterol would limit disease progression. However, they have not provided any animal models with lowered cholesterol diet to provide evidence to their claim. The authors should re-write the conclusion as no diet based experiments were performed in the study.

Several changes to the grammar and sentence structure are necessary throughout the manuscript.

Reviewer 3 Report

General comments

- LDL means already lipoprotein. Please correct "LDL lipoprotein" by "LDL"

- Take care about English writing: sometimes I see "h" for hour and sometimes "hrs". Numerous othe rmistakes have been made. Corrections should be done seriously. All along fth emanuscript, English is quite poor and some sentences should rewritten.

Results

- Figure 1c: what is the legend of y-axis ? It seems that it is not a concentration. The number of cells for which volum of culture medium ? I do not understand the meaning of *. What has been compared ?

- L119-129: this section is more in relation with the Material and Methods section.

- L130: what is the vitality of a cell line ?

- Fig 2b: the increase of MCF7 with hyperchol is higher than with LD of hypochol. A comparison of mean by statistics should give more information. If you compare the different means, the Student t test is not enough.

- L132-134: the authors cannot write that there are higher or lesser effects without a comparion of all the means.

- Fig 4a: the experimental results are compared to what ? is there any difference between the LD and HypoChol ? The means +/- SD seem to be the same.

- Fig 5: same comment than for Fig 4a, concerning CDKN1A

Discussion

- Numerous items of the discussion are not really in relation with the results. The authors do not use their own results to discuss them. The authors should take only their results and discuss them, specifically concerning the lipid rafts: cholesterol is not enough to explain change in lipid rafts.

- L263-265: the authors give some propsoitons for future studies but, why they did not do them and specifically concerning the mechanism of action of Chol ?

Material and Methods

- L297-302: What is the solvent of the LDL ?

- L310: why using 1% and 4% DMSO ? is it in relation with hypo- and hypercholesterolemia ?

- L324-334: the indicated volumes of chloroform/methanol should depend on volume of culture medium and of number of cells. This should be precised. The use of TLC is enough resolutive according to the amount of lipids and thus, according to the number of cells. This shoud be explain and precised. How the authors can be sure that lipids were also present in the second and third extracts ?

- L 372-375: is the Student's t test the most appropriate ? the use of ANOVA would be more efficient to compare all the results. The use of Dunnett test should also give different p value.

Reviewer 4 Report

This work deals with a subject of interest. However,  the present version can be improved in a number of aspects. First depth of research could be increased.

Regarding the presentation:

Figure 1. should be a supplemental figure.

Complete gel pictures with coomassie blue or another similar stain could be shown in the original images file.